# Immunodetection of Truncated Forms of the α6 Subunit of the nAChR in the Brain of Spinosad Resistant *Ceratitis capitata* Phenotypes

**DOI:** 10.3390/insects14110857

**Published:** 2023-11-04

**Authors:** Ana Guillem-Amat, Elena López-Errasquín, Irene García-Ricote, José Luis Barbero, Lucas Sánchez, Sergio Casas-Tintó, Félix Ortego

**Affiliations:** 1Centro de Investigaciones Biológicas Margaritas Salas, CSIC, 28040 Madrid, Spainortego@cib.csic.es (F.O.); 2Instituto Cajal, CSIC, 28002 Madrid, Spain

**Keywords:** medfly, α6 translation, protein location, resistance mechanism, insecticide resistant

## Abstract

**Simple Summary:**

The insecticide spinosad is widely used for the control of the Mediterranean fruit fly (medfly), *Ceratitis capitata*, in citrus crops in Spain. However, the sustainable use of this insecticide is compromised by the detection of spinosad resistant alleles in field populations, which may lead to control failures in the future. Mutations, generating premature stop codons at the gene coding for the target alpha 6 subunit of the nicotinic acetylcholine receptor of *C. capitata* (*Ccα6*), have been associated with spinosad resistance in both laboratory strains and field populations. In this work, we showed that full-length transcripts from individuals carrying wild-type isoforms of the gene are translated into Ccα6 proteins that locate in the membrane of the brain cells, while truncated transcripts from spinosad resistant strains could be translated into truncated Ccα6 that, for the most part, are unable to reach their expected functional destination in the membrane. We proposed that the difference of location of Ccα6 observed in spinosad resistant strains is probably determining its resistant phenotype. In addition, we provide a tool for immunodetection of truncated forms of Ccα6, that can be useful for resistance management programs.

**Abstract:**

The α6 subunit of the nicotinic acetylcholine receptor (nAChR) has been proposed as the target for spinosad in insects. Point mutations that result in premature stop codons in the *α6* gene of *Ceratitis capitata* flies have been previously associated with spinosad resistance, but it is unknown if these transcripts are translated and if so, what is the location of the putative truncated proteins. In this work, we produced a specific antibody against *C. capitata* α6 (Ccα6) and validated it by ELISA, Western blotting and immunofluorescence assays in brain tissues. The antibody detects both wild-type and truncated forms of Ccα6 in vivo, and the protein is located in the cell membrane of the brain of wild-type spinosad sensitive flies. On the contrary, the shortened transcripts present in resistant flies generate putative truncated proteins that, for the most part, fail to reach their final destination in the membrane of the cells and remain in the cytoplasm. The differences observed in the locations of wild-type and truncated α6 proteins are proposed to determine the susceptibility or resistance to spinosad.

## 1. Introduction

Insecticide resistance is one of the major threats against pest management in the world and a major concern for agriculture, since the number of resistance cases has increased exponentially over the past 60 years [1]. The Mediterranean fruit fly, *Ceratitis capitata*, one of the main pests of citrus and other fruits worldwide, is not an exception, and resistance to some of the insecticides used for its control has already been detected in field populations, as to malathion [2] and lambda-cyhalothrin [3]. Spinosad is one of the main insecticides used to fight medfly and, though field resistance has not yet arisen, resistant alleles have been detected at a low proportion [4]. Therefore, understanding of the mechanism leading to resistance is urgent to assure an effective resistance management [5].

Resistance to spinosad in *C. capitata* is mediated by point mutations in the nicotinic acetylcholine receptor (nAChR) α6 subunit gene (*Ccα6*), which lead to premature stop codons [4,6,7]. This gene comprises 12 exons with mutually exclusive isoforms containing alternative exon 3 (3a, 3b) and exon 8 (8a, 8b, and 8c) [6,8]. The wild-type insect α6 subunit contains an extracellular N-terminal domain with six loops, four transmembrane segments (TM1–4), a large intracellular linker (between TM3–4), and a C-terminal extracellular region [9,10,11] (Appendix A). In the laboratory strain JW-100s, selected for spinosad resistance, four different alleles carrying mutations in the *Ccα6* gene have been associated to the resistant phenotype: (i) *Ccα6^3aQ68*Δ3b-4^*, containing a Q68* mutation on exon 3a that generates a premature stop codon and a deletion of exons 3b and 4 in the gDNA, thus giving rise to transcripts that truncate at exon 3a and to transcripts lacking exons 3b and 4 that truncate in exon 5 due to a change in the reading frame (Appendix A); (ii) *Ccα6^3aQ68*-K352*^*, containing the mutations Q68* in exon 3a and K352* in exon 10, both generating premature stop codons, thus leading to transcripts truncated at exon 3a and to transcripts that carry exon 3b and truncate at exon 10 (Appendix A); (iii) *Ccα6^3aQ68*^*, containing the Q68* mutation that generates a premature stop codon on exon 3a, thus giving rise to transcripts truncated in exon 3a and to wild-type full-length transcripts with exon 3b; (iv) *Ccα6^3aAG>AT^*, containing a point mutation that changes the 5′ AG splicing site of exon 3a to AT, thus producing incomplete transcripts that skip exon 3a without altering the coding frame, and wild-type full-length transcripts with exon 3b [6,7]. Interestingly, the frequency of the two resistant alleles not expressing full-length Ccα6 isoforms (*Ccα6^3aQ68*-K352*^* and *Ccα6^3aQ68*-K352*^*) increased during the selection process, whereas those alleles that produced wild-type full-length transcripts with exon 3b (*Ccα6^3aQ68*^* and *Ccα6^3aAG>AT^*) were eliminated [6,7]. In Spanish field populations of medfly, two spinosad resistant alleles have also been identified: one carrying the mutation K352*, and the other producing transcripts that lack exons 5–11 caused by a deletion or a splicing alteration [4]. The expression of the resistant allele carrying mutation 3aQ68* in *Drosophila melanogaster* using the GAL4>UAS system was used to functionally validate the role of truncated isoforms of the α6 subunit in the resistant phenotype [7]. In addition, the generation of two isolines formed by individuals homozygous for the alleles *Ccα6^3aQ68*-K352*^* (isoline Q68*-K352*, resistance ratio (RR) = 3062) or *Ccα6^3aQ68*Δ3b-4^* (isoline Q68*, RR = 3139) has allowed the observation that these resistant alleles not expressing full-length Ccα6 isoforms endure a cost of behavioral fitness traits [7]. Spinosad resistance due to mutations and/or exon skyping events that result in premature stop codons in the *α6* gene have also been reported in different insect species [12,13,14]. However, despite it being assumed that truncated mRNA products presumably result in non-functional truncated proteins, whether they are truly translated and what is the location of these putative truncated-proteins in vivo still needs to be determined. 

With this aim, in this work we conducted the following: (i) produced an antibody for in vivo recognition of both wild-type and truncated forms of Ccα6; (ii) tested the utility of the antibody to recognize α6 subunits by immunofluorescence assays performed with wild-type and *α6-dsRNA* expressing strains of *D. melanogaster*; (iii) analyzed the location of Ccα6 in brains of control and spinosad resistant strains of *C. capitata*; (iv) discussed whether differences in Ccα6 location between resistant and susceptible individuals help to explain their association with the spinosad resistance phenotype. 

## 2. Materials and Methods

### 2.1. Ceratitis capitata and Drosophila melanogaster Strains

The wild-type laboratory strain of *C. capitata* (C) was established from medfly individuals collected from non-treated experimental fields of the Instituto Valenciano de Investigaciones Agrarias (IVIA, Valencia, Spain) in 2001, and reared in the laboratory without any exposure to insecticides for about 200 generations. The susceptibility of the wild-type strain to spinosad has been determined for both feeding (LC_50_ = 0.58 ppm of spinosad in the diet) and topical (LD_50_ = 0.10–0.12 μg of spinosad/g of insect) application [6,7]. The Q68*-K352* (resistance ratio (RR) = 3062) and Q68* (RR = 3139) spinosad resistant isolines, homozygous for *Ccα6^3aQ68*-K352*^* and *Ccα6^3aQ68*Δ3b-4^* alleles respectively, were derived from the spinosad resistant laboratory strain JW-100s (RR = 1794) [6,7], and maintained under regular selection in the laboratory.

The following *D. melanogaster* lines purchased from Bloomingtong Stock Center were used in the control experiments with different purposes: stock Dm 84665: *TI{2A-GAL4}nAChRalpha6[2A-GAL4]/CyO*, which expresses GAL4 under the control of nAChRalpha6 regulation; stock Dm 25835: *y [1]v [1];P{y[+t7.7]v[+t1.8] = TriP.JF01853}attP2*, which expresses dsRNA for RNAi of nAChRalpha6 (FBgn0032151) under UAS control in the VALIUM10 vector; stock Dm 57818: *y [1]v [1];P{y[+t7.7]v[+t1.8] = TriP.HMJ21826}attP40/CyO*, which expresses dsRNA for RNAi of nAChRalpha6 (FBgn0032151) under UAS control in the VALIUM20 vector; stock BL 52676: *w [1118]; P{y[+t7.7] w[+mC] = GMR27E08-lexA}attP40*, which expresses lexA under the control of DNA sequences in or near elav; stock BL 66545: *w[*]; P{w[+mC] = LexAop-CD8-GFP-2A-CD8-GFP}2; TM2/TM6B, Tb [1]*, which expresses membrane-localized CD8-GFP under the control of LexAop. All of them were kept in the laboratory without exposure to insecticides. Flies were kept in standard fly food at 25 °C, unless otherwise indicated. To analyze the effectivity of the dsRNAs for RNAi, males from the stocks carrying the dsRNA of nAChRalpha6 (Dm 25835 and Dm 57818) were crossed with virgin females from the stock expressing GAL4 under the control of nAChRalpha6 promoter (Dm 84665). Pooled crosses were established to obtain the genotypes *nAChRα6-gal4 > UAS-nAChRα6^RNAi−25835^* [α6-dsRNA (1)] and *nAChRα6-gal4 > UAS-nAChRα6^RNAi−57818^* [*α6-dsRNA* (2)], both producing the nAChRalpha6 dsRNA, distinguishable by not showing the CyO phenotype but the wild-type. Likewise, BL 52676 and BL 66545 flies were crossed to generate *w; elav-LexA>LexAop-CD8GFP* F1 genotypes which express membrane-localized CD8-GFP and wild-type *Dα6*. The resulting F1 flies were collected and used for immunofluorescence assays (see Section 2.4).

### 2.2. Heterologous Expression of Truncated Isoform Ccα6^3aQ68*^

The truncated isoform Ccα6^3aQ68*^ (N-terminal fragment of Ccα6, corresponding to exons 1, 2, and 3a truncated at residue Q68 (see Appendix A), was generated by molecular cloning and heterologous expression. The N-terminal of *Ccα6* was amplified by colony PCR with *Escherichia coli* carrying isoform *Ccα6^3a8b^* cloned in a pGEM-T-easy vector (Promega Corporation, Madison, USA). The isoform was amplified in a volume of 250 μL using 0.4 μM of forward alpha6-NtermF (5′-CGCATATGGACCCGTCGCTGTTTG) and reverse alpha6-NtermR (5′-GCGTCGACCTAGGGCACGTACAGAC) oligonucleotides, 2 U of AmpliTaq Gold DNA Polymerase (Thermo Fisher Scientific, Austin, USA), 10× PCR Buffer II, 1.5 mM MgCl_2_, 0.2 mM dNTPs (Thermo Fisher Scientific, Austin, USA), and a colony carrying the specific isoform. PCR conditions were as follows: an initial denaturation step at 95 °C for 10 min; 40 cycles of 95 °C for 30 s, 65 °C for 30 s, and 72 °C for 40 s, with a final step of 72 °C for 10 min for full extension. PCR products were analyzed by electrophoresis on 1.2% agarose gel (Agarose D2, Conda Pronadisa, Madrid, Spain). The bands of interest were purified from the gel with NZYGelpure (NZYTech, Portugal) following the manufacturer’s instructions, and the Ccα6^3AQ68*^region was then sequenced (Secugen, Madrid, Spain). PCR products were cloned into the pGEM-T-easy vector following the manufacturer’s instructions. Transformation of DH5α cells (NZYTech, Lisbon, Portugal) was performed using a standard heat shock protocol and cultures were established. Plasmids were purified (High Pure Plasmid Isolation Kit, Roche, Germany), eluted in 100 µL and sequenced (Secugen, Madrid, Spain). Clones were digested with *Nde*I and *Sal*I (NZYTech, Portugal) and fragment-size was checked by electrophoresis on 1.2% agarose gel. Digested fragments were cloned into the pET12a following the manufacturer’s instructions and DH5α were transformed by heat shock, cultures established, and fragment-size checked by *Nde*I and *Sal*I digestion and electrophoresis. Clones for Ccα6^3AQ68*^ were transformed into BL21(DE3) cells (NZYTech, Portugal) by heat shock, cultures established, and plasmids purified, while the presence of the fragment was checked by digestion and electrophoresis.

Transformed BL21(DE3) cells were grown in LB medium with ampicillin (150 mg/mL) at 37 °C and 200 rpm for 6 h, IPTG was then added to a final concentration of 2 mM and cultures were cultivated over-night. Cultures without IPTG were used as non-induction controls. Cultures were centrifuged at 4000 rpm for 10 min at 4 °C and pellets were resuspended and sonicated for 2 min. Tubes were centrifuged at 10,000 rpm for 10 min at 4 °C. Soluble and non-soluble fractions were separated. SDS-PAGE in a 4–20% polyacrylamide gel (Mini-PROTEAN^®^ TGX^TM^ Precast Protein Gels, Bio-Rad, Spain) was performed to analyze the protein extracts. Once the induction of the expression was checked, preparative gels of electrophoresis were performed to separate the specific band, which was then homogenized (IKA ULTRA-TURRAX homogenizer T-18, Merck, Germany) and centrifuged at 4000 rpm for 10 min twice, discarding the pellet. Supernatant was lyophilized over-night. It was then resuspended in distilled water and dialyzed for 90 min. Purified Ccα6^3aQ68*^ was sequenced (Protein Chemistry Service, CIB Margarita Salas, Madrid, Spain) to obtain the N-terminal sequence.

### 2.3. Generation and Validation of the Anti-Ccα6 Antibody

The peptide Ccα6 (N-term KESCQGPHEKRLLNHLLSTYNTLER C-term) was designed from the sequence of *Ccα6*. It was synthetized, purified by Hplc, and joined to KLH (keyhole limpet hemocyanin) at the Proteomics’ department of the National Center for Biotechnology (Madrid, Spain).

Immunization was performed on female New Zealand rabbits at the Animal Facility of the CIB Margarita Salas. The purified synthetic peptide Ccα6 was used as antigen at a concentration of 1 mg/mL. It was emulsified with complete Freund’s adjuvant (1:1) for the first injection, and with incomplete Freund’s adjuvant (1:1) for the second and third injections. Injections were spaced three weeks apart. A pre-immune blood sample was extracted before the first immunization as a control. Blood samples were centrifuged at 4 °C and 4000 rpm for 10 min, and supernatants were aliquoted and stored at −20 °C. The anti-Ccα6 antibody was purified from the serum sample using HiTrap NHS-activated HP columns (Cytiva, Sweden) coupled with the purified synthetic peptide Ccα6, following purification and coupling procedures using a syringe described in the manufacturer’s Antibody Purification Handbook. Elution was achieved by lowering the pH to 2.5.

The titer of the antibody in the serum and after purification was analyzed by an ELISA assay using the antigen used for its generation. It was performed in flat bottom clear polystyrene 96 well plates (Nunc-Inmuno^TM^ MicroWell^TM^ 96 well solid plates, Sigma-Aldrich, St. Loius, MO, USA) and wells were washed with distilled water between steps. First, wells were coated with 5 µg/mL of Ccα6 antigen in Na_2_CO_3_ (0.1 M, pH 9.6) and incubated over-night at 37 °C. Residual action spots were blocked with BSA 1% in PBS for 1 h at 37 °C. The antibody was diluted in BSA 0.5%, added to the corresponding wells, and incubated for 2 h at 37 °C. A secondary antibody anti-rabbit conjugated to HRP (1:2000; Dako Products, Agilent, Santa Clara, CA, USA) was added and incubated for 1 h at 37 °C. A H_2_O_2_ and OPD (o-phenyenediamine dihydrochloride, Sigma-Aldrich, St. Louis, MO, USA) 1:2000 solution was added and incubated for 5–20 min. The reaction was stopped by adding H_2_CO_4_ 3M. Absorbance was measured at 492 nm. Serial dilutions of pre-immune and immune serums and purified anti-Ccα6_antibody were tested by duplicate, and wells incubated with BSA 0.5% were used as controls (Appendix A).

The capacity of the immune serum and purified anti-Ccα6 antibody to recognize the truncated isoforms of Ccα6 was validated by Western blotting. The truncated Ccα6^3aQ68*^ was denaturalized, separated using 4–20% Mini-PROTEAN TGX precast protein gels (Bio-Rad, Hercules, CA, USA), and transferred onto nitrocellulose membrane. The membrane was blocked for 1 h at room temperature in PBST (0.05% Tween-20 in PBS) containing 1% BSA. Primary antibody incubation was performed with immune serum or purified anti-Ccα6 antibody overnight at 4 °C at 1:100 or 1:50 dilution, respectively. After washing in PBST, membranes were incubated for 1 h at room temperature with a polyclonal goat anti-rabbit HRP-conjugated secondary antibody (Dako Products, Agilent, USA). Finally, membranes were washed in PBST and developed using the ECL Western Blotting detection reagents (Sigma-Aldrich, USA) (Appendix A).

### 2.4. Immunofluorescence Assays

Adult brains were dissected and fixed with 4% formaldehyde in phosphate-buffered saline for 35 min. Next, samples were washed 3 × 15 min with PBS + 0.1% triton (PBT) and blocked overnight with PBS + 0.1% triton + BSA 5% (PBTX). Then, they were incubated overnight with the primary antibody diluted in PBTX, washed 3 × 15 min in PBT, incubated with secondary antibodies diluted in PBTX for 2 h, mounted with Vectashield mounting medium (Vector Laboratories, Ref H-1200) with DAPI in microscopy slides, and stored at 4 °C. The primary antibodies used were anti-Ccα6 antibody serum (1:500) and purified anti-Ccα6 antibody (1:50) to recognize the N-terminal region of the α6-subunit of the receptor (see Section 2.2) and anti-elav rat (1:50; DSHB AB_528218) to recognize the nucleus of the neuron. The secondary antibodies used were anti-rabbit Alexa 568 (1:500; Thermo Fisher, Ref A11011) and anti-rat Alexa 488 (1:500; Thermo Fisher, Ref A 11006). Samples were analyzed by confocal microscopy (LEICA TCS SP5). We used 20×, 40× and 63× objectives and images were acquired every 1 µm. Fluorescent signal intensity was measured using the software ImageJ 1.53t. Single stacks of confocal images were analyzed using the Analyze->Plot Profile tool. Briefly, we drew 6 single lanes that cross a single cell per image, or a square of the same size for every image on the regions of interest, and we obtained the histogram of pixel intensity (from 0 to 255) and the distance. Intensity signal was statistically analyzed using one-way ANOVA followed by Tukey´s multiple post hoc test in GraphPad Prism 9.

## 3. Results

### 3.1. Evaluation of the Anti-Ccα6 Antibody

Rabbit immunization with a peptide corresponding to the N-terminal domain of the Ccα6 protein produced an immune serum containing the anti-Ccα6 antibody, which was further purified by affinity chromatography. Both immune serum and purified anti- Ccα6 antibody were validated by ELISA assay (Appendix A) and Western blotting (Appendix A).

The capacity of the immune serum and purified anti-Ccα6 antibody to detect in situ the α6 subunit of the nAChR was determined by immunohistochemistry with different strains of *D. melanogaster* (Figure 1 and Appendix A). A specific and intense fluorescent signal was observed in brain cells from different adult flies expressing wild-type *Dα6* (Figure 1A,D and Appendix A). The α6 signal co-localized with the expression of the membrane-bound reporter CD8-GFP (Appendix A), indicating that the anti-Ccα6 antibody specifically binds to the membrane of the neurons. The quantification of the α6 signal through single brain cells resulted in two peripheral peaks corresponding to the expected location in the cell membrane (Figure 1D´). However, only a weak background signal was detected in individuals expressing *α6-dsRNA* (Figure 1B,C,E,F), which did not show a distinct peripheral distribution in the cell when it was quantified (Figure 1E´,F´). Moreover, the signal corresponding to the anti-Ccα6 antibody was detected in higher intensity in control samples than in α6-dsRNA (1) (Figure 1B,E) and α6-dsRNA (2) individuals (Figure 1C,F). Taken together, these results indicate the specificity of the anti-Ccα6 antibody to recognize the α6 subunit of the nAChR of *D. melanogaster*, but not other nAChR subunits, as using RNAi to reduce *Ccα6* expression coincides with a significant reduction (α6-dsRNA (1)) or suppression (α6-dsRNA (2)) in the antibody staining (Figure 1G).

### 3.2. Presence of Ccα6 in the Brains of Ceratitis capitata

Immunohistochemistry experiments performed with anti-Ccα6 antibody serum (Figure 2A,B) and purified anti-Ccα6 antibody (Figure 3A–D) suggested that Ccα6 is expressed along all the brain areas of *C. capitata* wild-type individuals. As already observed in the brains of *D. melanogaster*, this protein is naturally located in the membrane of the brain cells of individuals carrying wild-type isoforms of the gene, since the Ccα6 signal is mostly located at the peripheral area of the cells (Figure 3B,E) and neither overlaps with the DAPI signal of the genetic material (Figure 3A) nor with the neurons’ nucleus (Figure 3C). The merging of all three signals can be observed in Figure 3D. Moreover, the quantification of the Ccα6 signal through single brain cells resulted in two peaks that correspond to the membrane (Figure 3E’). However, the fluorescent signal referring to the position of the Ccα6 protein was more diffused in both resistant isolines Q68* and Q68*-K352* (Figure 3F,G), and its quantification through single brain cells did not result in defined peaks corresponding with the membranes (Figure 3F´,G´), compared to wild-type C strain (Figure 3E,E’). In addition, a significant reduction in the fluorescent signal was also observed in both resistant strains (about 22 and 37% for Q68* and Q68*-K352*, respectively) compared to the wild-type (Figure 3H). This result suggests that truncated transcripts could be translated into truncated proteins that, for the most part, are unable to reach their expected functional destination in the membrane. The different observed signal in the resistant and the susceptible strains indicates the existence of a difference in the location of the protein associated to the phenotype.

## 4. Discussion

In this work we produced and evaluated an antibody against the α6 subunit of the nAChR of *C. capitata* with the aim of detecting if the location of this subunit in the brain of the flies could be involved in spinosad resistance. Several works confirmed the role of the α6 subunit of the nAChR as the target of spinosad in dipterans, both in the model organism *D. melanogaster* [15,16] and in the pest we are dealing with in this work, *C. capitata* [7]. In addition, recent publications stated that spinosyns may exclusively act on α6 homomeric nAChRs, which was observed for both spinosad [17] and spinetoram [18]. However, to our knowledge very few works have focused on the location of this protein in the insects’ nervous system [19], while the capability of truncated transcripts to be translated into proteins in vivo remains unknown. Our results confirmed that *C. capitata* and *D. melanogaster* strains carrying wild-type versions of *α6* gene give rise to full-length transcripts that are translated into proteins that reach their expected destination in the cellular membrane, as already observed previously for other nAChR subunits [20]. Furthermore, we reported for the first time that truncated transcripts of *Ccα6* present in the resistant isolines of *C. capitata* Q68* and Q68*-K352* are also translated. Remarkably, the fact that the fluorescent signal in these strains is not located specifically in the membrane of the brain cells suggests that truncated transcripts are expressing truncated proteins that cannot carry their expected function as transmembrane proteins. In addition, our results showing a lower Ccα6 fluorescent signal in the resistant isolines are in line with those obtained in a previous work showing that the expression of different exonic regions of the α6 subunit was lower in the resistant isolines Q68* and Q68*-K352* than in the susceptible strain C; this difference being statistically significant in the case of Q68*-K352* [7]. Nevertheless, lower translation rates of mRNA transcripts and/or higher degradation of truncated proteins may also occur. Also, we cannot discard that isoforms lacking exons 3b and 4 and truncating at exon 5 (isoline Q68*) have a better accessibility to the antibody than transcripts carrying exon 3b and truncating at exon 10 (isoline Q68*-K352*). Previous works hypothesized that *α6* truncated transcripts and/or putative truncated proteins may regulate the expression of full-length nAChR transcripts [21,22], lead to compensatory changes in the expression levels of other nAChR subunits [23], or modulate cholinergic synaptic transmission [24]. However, further work will be required to establish if the α6 truncated proteins identified in the cytoplasm and membrane of the brain cells have a physiological function or whether they are in transit to be degraded.

Considering that an alteration in the location of a membrane protein can affect its function, we hypothesize that this could be the reason why alleles *Ccα6^3aQ68*-K352*^* and *Ccα6^3aQ68*Δ3b-4^*, that give rise to truncated transcripts, confer a resistant phenotype to the individuals that carry them. Nevertheless, the fact that truncated forms are lacking the epitope to which spinosad is supposed to bind suggests that, even a small proportion of them are apparently reaching the membrane, with the resultant phenotype being resistant. Thus, we cannot discard the view that the resistant phenotype is not exclusively due to a change in the location of the protein but rather because of a change in the protein structure. We should outline that, despite these hypotheses, the reason why individuals homozygous for *Ccα6^3aQ68*^* or *Ccα6^3aAG>AT^* alleles are resistant remains unknown, as they affect transcripts containing exon 3a but do generate wild-type, full-length *Ccα6* transcripts with exon 3b. Guillem-Amat et al. [7] demonstrated that the expression of wild-type isoforms carrying exon 3a (*Ccα6^3a8b^*) and exon 3b (*Ccα6^3b8a^*) in *D. melanogaster* restored susceptibility to spinosad in a α6-deficient strain, indicating that medfly *Ccα6^3b8a^* isoform is enough to mediate spinosad toxicity. Therefore, alternative hypotheses that explain the resistant phenotype conferred by *Ccα6^3aQ68*^* and *Ccα6^3aAG>AT^* alleles, both encoding *Ccα6^3b8a^* isoform, need to be tested, as for instance the hypothesis that truncated 3a isoforms may interfere with the correct assembly of nAChR receptors in medfly, previously formulated by Ureña et al. [6]. The obtention of specific antibodies for 3a and 3b isoforms would also be an interesting tool, since immunohistochemical assays performed with them could provide us with information regarding whether or not there are differences in the location of both isoforms.

Our work also gives some light on the topic of α6 subunit location in the nervous system. Different nAChR subunits were found in the developmental stages of embryo, larva, pupa, and adult of some insect species [25]. In adult individuals of *D. melanogaster*, subunits Dα1, Dα3, Dβ1, Dβ2, Dα7 were mainly localized in the optic lobe, spread in the neuropils, the lamina, the medulla, the lobula, and the lobula plate [26,27,28]. Dα1 and Dβ1 were also found in the protocerebrum, the deutocerebrum, and the thoracic ganglion [29]. Regarding subunit Dα6, it was localized in previous works within the neuropil of the ventral lateral neurons (LNv) dendritic arbor, as does subunit Dα1 [19]. Our results show that Ccα6 translation occurs along all the brain areas of medfly, meaning that the expression of this protein is ubiquitous. In addition, this work also demonstrates that at a subcellular level Ccα6 is located in the plasma membrane, as expected for a membrane receptor and already observed for some nAChR—concretely nAChRβ1 [20]. Further knowledge about the location of Ccα6 could help to understand the biology and the relationship of this specific subunit with spinosad resistance in this pest.

## Figures and Tables

**Figure 1 insects-14-00857-f001:**
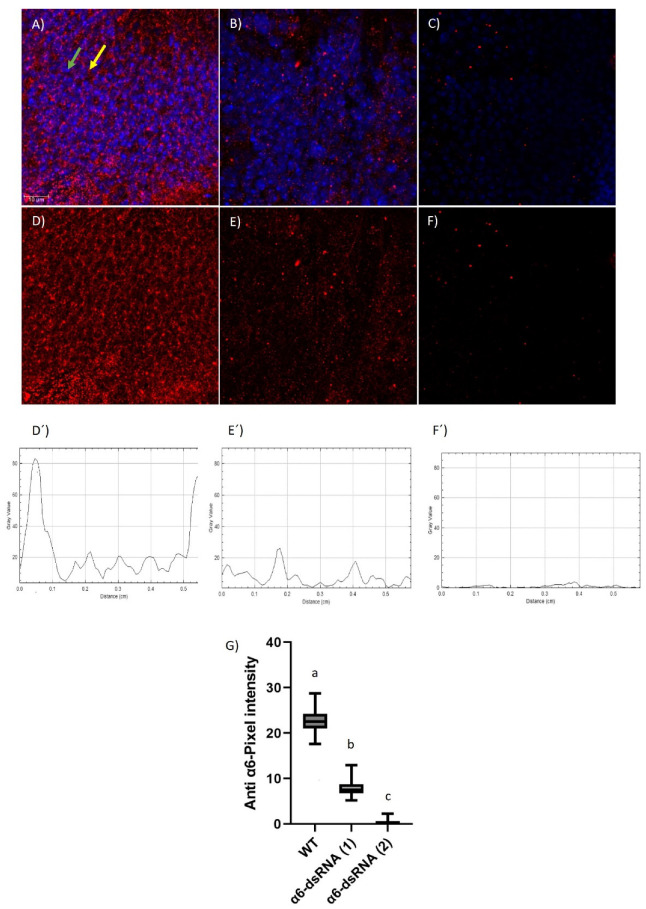
Location of α6 subunit of the nAChR (red) and the genetic material (blue, DAPI) in adult brain cells of the following *Drosophila melanogaster* phenotypes (genotypes): (WT) wild-type individuals for α6 expression from Dm 84665 strain (*nAChRα6-gal4*) (**A**,**D**); α6-dsRNA (1) individuals coming from the cross of Dm 84665 strain with the UAS line Dm 25835 (*nAChRα6-gal4 > UAS-nAChRα6^RNAi-25835^*) (**B**,**E**); and *α6-dsRNA* (2) individuals coming from the cross of Dm 84665 strain with the UAS line Dm 57818 (*nAChRα6-gal4 > UAS-nAChRα6^RNAi-57818^*) (C and F). In all pairs, both images belong to the same picture showing both Dα6 and DAPI simultaneously (**A**–**C**) or only Dα6 (**D**–**F**). Samples were analyzed by confocal microscopy (20×, 40× and 63× objectives; images were acquired every 1 µm). Arrows spotting to a: cell nucleus (green), α6 protein (yellow). Fluorescent nAChR (red) signal intensity was quantified in brain cells (5–7 µm) from WT (**D’**) α6-dsRNA (1) (**E’**) and α6-dsRNA (2) (**F’**) individuals. Average pixel intensity of Ccα6 signal per image was measured (n = 6) and statistically analyzed (different letters over the error bars indicate statistically significant differences, *p* < 0.05) using one-way ANOVA followed by Tukey’s multiple post hoc test (**G**).

**Figure 2 insects-14-00857-f002:**
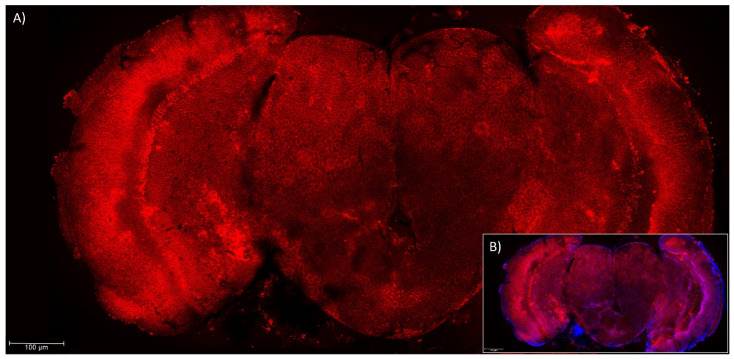
Location of Ccα6 (red) and the genetic material (blue, DAPI) in the brain of a wild-type individual of *Ceratitis capitata*, belonging to C strain. Both images belong to the same picture showing only Ccα6 (**A**) or both Ccα6 and DAPI simultaneously (**B**). Samples were analyzed by confocal microscopy (10×).

**Figure 3 insects-14-00857-f003:**
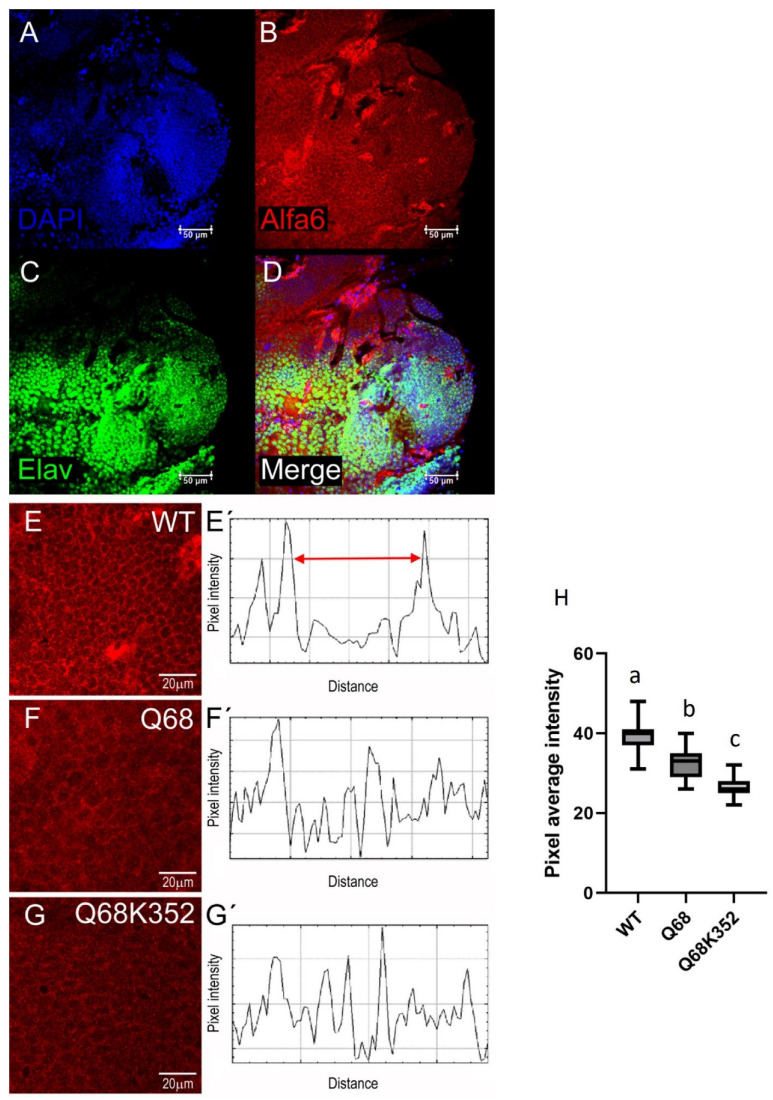
Location in wild-type *Ceratitis capitata*’s brain cells of: (**A**) genetic material (blue, DAPI); (**B**) Ccα6 (red); and (**C**) neuron’s nucleus (green). (**D**) Shows the merging of DAPI, Ccα6, and neuron nucleus signals. Ccα6 location and fluorescence intensity in flies from the strains: WT) wild-type phenotype for Ccα6 expression (**E**,**E′**); Q68*) homozygous for the allele *Ccα6^3aQ68*Δ3b-4^* and with a spinosad resistant phenotype (**F**,**F′**); Q68*-K352*) homozygous for the allele *Ccα6^3aQ68*-K352*^* and with a spinosad resistant phenotype (**G**,**G′**). Samples analyzed by confocal microscopy (20×, 40× and 63× objectives; images were acquired every 1 µm) corresponded to randomly chosen adult females. Fluorescent nAChR (red) signal intensity was quantified in single brain cells (5–7 µm) from WT (**E′**) Q68* (**F′**) and Q68*-K352* (**G′**) individuals. Average pixel intensity of Ccα6 signal per image was measured (n = 6) and statistically analyzed (different letters over the error bars indicate statistically significant differences, *p* < 0.05) using one-way ANOVA followed by Tukey´s multiple post hoc test (**H**).

## Data Availability

The data presented in this study are available in the article and Appendix A.

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
