# Peer review of "Immunodetection of Truncated Forms of the α6 Subunit of the nAChR in the Brain of Spinosad Resistant Ceratitis capitata Phenotypes"

_insects, 2023, doi:10.3390/insects14110857_

Round 1
Reviewer 1 Report (Previous Reviewer 2)
Comments and Suggestions for Authors
This is an elegant piece of work generating a new tool that could help on the study of the mechanisms behind spinosad resistance in Ceratitis capitata. This reviewer would like to thank the authors for addressing all the comments made in the previous revision.
Author Response
Authors thank this reviewer for the review performed.
Reviewer 2 Report (Previous Reviewer 4)
Comments and Suggestions for Authors
The authors have satisfactorily addressed the corrections and suggestions indicated.
Author Response
Authors thank this reviewer for the review performed.
This manuscript is a resubmission of an earlier submission. The following is a list of the peer review reports and author responses from that submission.
Round 1
Reviewer 1 Report
Comments and Suggestions for Authors
Please see attached file

Reviewer 2 Report
Comments and Suggestions for Authors
The study entitled “Immunodetection of truncated forms of the α6 subunit of the nAChR in the brain of spinosad resistant Ceratitis capitata phenotypes”, by Guillem-Amat and coauthors, generates an antibody to detect nAChR α6 subunits, which represents a very interesting and useful tool for the study of this subunit in insect pest species, given its importance in spinosad resistance. The authors show the utility of the antibody on the detection of nAChR α6 subunits by immunohistochemistry in Drosophila and C. capitata brains, and a reduction of the levels in C. capitata strains expressing truncated forms of nAChR α6 subunits. On top of the high utility of this tool stated above, the study is very elegantly executed, technically impeccable, and nicely written. For all these reasons, this reviewer recommends publication of this work after some corrections included in the minor comments described below.
Minor comments:
- In Material and methods, section 2.3, the reference number for the Vectashield and secondary antibody Alexa 568 should be included to facilitate repeatability.
- Scheme1 and Supplementary figure1 are the same.
- Scheme1, section b: the allele is supposed to form two type of isoforms, but only one is represented in the scheme.
- Line 78: Ceratitis capitata should be in italic.
- Scheme2 and Supplementary Figure 2 are the same.
- Line 168: “wild type Ccα6” should be changed for “wild type Dα6”
- Figure1: To facilitate the understanding of the figure the authors should write the genotypes in the figure following the nomenclature widely used for Drosophila genotypes: nAChRα6-gal4; nAChRα6-gal4 > UAS-nAChRα6RNAi-25835; and nAChRα6-gal4 > UAS-nAChRα6RNAi-84665. Furthermore, the authors should add correspondence between staining and colour in one of the panels of the figure (nAChRα6, DAPI)
- Figure 1: This reviewer thinks the interpretation and clarity of subcellular localization of alpha6 protein would benefit from a picture in higher magnification, if possible.
- Figure legend of Figure1: instead of referring the fly lines by their number, the authors should write the genotypes of the flies, as per the previous comment. This will help on understanding what is expressed in each panel.
- Line 173: quantitative comparisons between the expression of the two RNAi lines should be based on quantification of fluorescence intensity of Dα6.
- Lines 174-177: this reviewer thinks that the statement about the specificity of the antibody for α6 would be better supported with and align to proof the region of the antibody is specific for α6 and not conserved among other nAchRs, and/or including a literature reference where this is shown.
- Line 181-182: the description about which panels are red and which ones are red and blue in the figure legend does not correspond with the figure panels cited. Also, the authors should change “red” and “blue” for “Dα6” and “DAPI”.
- Line 184: the α6 products of the flies expressing dsRNA are not proven neither expected to be truncated; this expression should be changed. Furthermore, the protein detected is not Ccα6 but Dmα6.
- Line 183: yellow arrow indicates α6 protein, not the membrane.
- Figure 3: This reviewer thinks that, although the difference on the levels of α6 protein are apparent in the pictures, measurements of the fluorescence intensity would quantitatively proof the statements made.
- Line 192-196: This reviewer thinks that the main conclusion should be that the presence of the protein is much lower in the resistant isolines (quantification of fluorescence intensity would proof that), probably due to lower translation rates of mRNA transcripts and/or higher degradation of truncated proteins. Without co-staining with a membrane marker and the current pictures, this reviewer thinks there is no proof the resistant isolines analysed clearly show a difference on the location of the truncated proteins compared to wild type α6.
- Figure3: For easier understanding of the figure: add genotype in the figure, similar to what has been recommended for Figure 1. Moreover, add correspondence between staining and colour in one of the panels of the figure (α6, DAPI).
- Discussion: This referee wonders whether this antibody could be used for the detection of α6 in other insect species, as well as in Ceratitis capitata and Drosophila melanogaster. This would be very beneficial to study the role of α6 in other pest species where spinosad resistance is a challenge. A bit of discussion in this sense could be added in the discussion, as well as an alignment of the protein sequence of α6 of several insect species, using the region where the antibody binds and surroundings, to support or discourage this possibility.
Reviewer 3 Report
Comments and Suggestions for Authors
Resistance to spinosad in pests is mediated by point mutations in the nicotinic acetylcholine receptor (nAChR) α6 subunit gene, which lead to premature stop codons. However, it is un-known if these transcripts are translated and if so, what is the location of the putative truncated proteins. In this paper, the shortened transcripts were proved that present in resistant flies, generated putative truncated proteins but were unable to reach their final destination in the membrane of the cells and remain in the cytoplasm. I suggest the paper might be made acceptable pending revisions based on the comments below.
1, In materials and methods, the resistance level of Ccα63aQ68*Δ3b-4 or Ccα63aQ68*-K352* homozygous to spinosad should be introduced.
2, The fluorescent signal referred to the position of the Ccα6 protein was lower in both resistant isolines, Q68* and Q68*-K352*. Is this caused by a reduction in the transcription level of Ccα6 or by the translation process?
3, The Ccα63aQ68*-K352* strain can generate wild-type full-length Ccα6 transcripts with exon 3b. Whether the expression of various transcripts of Ccα6 in Ccα63aQ68*-K352* strain has been measured? Which transcript of Ccα6 is the most dominant in Ccα63aQ68*-K352* strain?
4, In figure 3, The Ccα6 protein of Ccα63aQ68*Δ3b-4 strain truncates in exon 5 due to a change in the reading frame How, the Ccα63aQ68*-K352* strain can generate wild-type full-length Ccα6 transcripts with exon 3b. Why was fluorescent signal of b and e (Ccα63aQ68*Δ3b-4) high than c and f (Ccα63aQ68*-K352*)?
5, line 82, “Ccα63aQ68*Δ3b-4” should be changed to “Ccα63aQ68*Δ3b-4”
6, line 84, “Ccα63aQ68*-K352” should be changed to “Ccα63aQ68*-K352”
Reviewer 4 Report
Comments and Suggestions for Authors
My only concern is the selection process of the wild-type and resistant strains of C. capitata.
Line 93-94. For how many generations the wild-type strain was bred in the laboratory? Did you perform bioassays with Spinosad to verify the susceptibility of this strain?
Line 97. What is the resistance level to Spinosad of the resistant lines of C. capitata?
Line 145. Adult brains were dissected and fixed with 4% formaldehyde in phosphate-buffered 145 saline for 20 min. What sex did you use?